# Discrepancies between Radiology Specialists and Residents in Fracture Detection from Musculoskeletal Radiographs

**DOI:** 10.3390/diagnostics13203207

**Published:** 2023-10-13

**Authors:** Jarno T. Huhtanen, Mikko Nyman, Roberto Blanco Sequeiros, Seppo K. Koskinen, Tomi K. Pudas, Sami Kajander, Pekka Niemi, Eliisa Löyttyniemi, Hannu J. Aronen, Jussi Hirvonen

**Affiliations:** 1Faculty of Health and Well-Being, Turku University of Applied Sciences, 20520 Turku, Finland; 2Department of Radiology, University of Turku, 20014 Turku, Finland; sami.kajander@utu.fi (S.K.); pekka.niemi@utu.fi (P.N.); 3Department of Radiology, Turku University Hospital, University of Turku, 20014 Turku, Finland; mikko.nyman@tyks.fi (M.N.); roberto.blanco@tyks.fi (R.B.S.); hannu.aronen@utu.fi (H.J.A.); jussi.hirvonen@utu.fi (J.H.); 4Terveystalo Inc., Jaakonkatu 3, 00100 Helsinki, Finland; seppo.koskinen@terveystalo.com (S.K.K.); tomi.pudas@terveystalo.com (T.K.P.); 5Department of Biostatistics, University of Turku, 20014 Turku, Finland; eliisa.loyttyniemi@utu.fi; 6Department of Radiology, Faculty of Medicine and Health Technology, Tampere University Hospital, Tampere University, 33100 Tampere, Finland

**Keywords:** competence, image interpretation, interpretation errors, discrepancy

## Abstract

(1) Background: The aim of this study was to compare the competence in appendicular trauma radiograph image interpretation between radiology specialists and residents. (2) Methods: In this multicenter retrospective cohort study, we collected radiology reports from radiology specialists (N = 506) and residents (N = 500) during 2018–2021. As a reference standard, we used the consensus of two subspecialty-level musculoskeletal (MSK) radiologists, who reviewed all original reports. (3) Results: A total of 1006 radiograph reports were reviewed by the two subspecialty-level MSK radiologists. Of the 1006 radiographs, 41% were abnormal. In total, 67 radiographic findings were missed (6.7%) and 32 findings were overcalled (3.2%) in the original reports. The sensitivity, specificity, positive predictive value, and negative predictive value were 0.86, 0.92, 0.88, and 0.91, respectively. There were no statistically significant differences between radiology specialists’ and residents’ competence in interpretation (*p* = 0.44). However, radiology specialists reported more subtle cases than residents did (*p* = 0.04). There were no statistically significant differences between errors made in the morning, evening, or night shifts (*p* = 0.57). (4) Conclusions: This study found a lack of major discrepancies between radiology specialists and residents in radiograph interpretation, although there were differences between MSK regions and in subtle or obvious radiographic findings. In addition, missed findings found in this study often affected patient treatment. Finally, there are MSK regions where the sensitivity or specificity is below 90%, and these should raise concerns and highlight the need for double reading and should be taken into consideration in radiology education.

## 1. Introduction

Health care is based on high-quality patient treatment, and to ensure this quality, the competence of health-care professionals needs to be systematically evaluated [1]. In medical imaging, the radiological report plays an important role in patient treatment [2] and helps general practitioners treating the patient. Radiographs are important in evaluating patients with upper- or lower-extremity trauma [3,4]. Thus, the radiology report based on radiographs has an important role in patient treatment.

Extremity fractures are the second-most missed diagnosis when reporting on radiographs [5]. This is especially relevant now that increased cross-sectional imaging represents a growing proportion of the teaching material during radiology residency training. Missed findings in radiographs may result in several complications for the patient [6]. Identifying mistakes made in radiograph interpretation is an important way to improve interpretation competence [7]. Up to 80% of diagnostic errors in radiology are classified as perceptual errors where the abnormal finding is not seen [2,8]. These errors are more frequent during evening and nighttime [9,10,11]. In skeletal radiology, most of malpractice claims towards radiologists are related to errors in fracture interpretation [12,13,14].

In summary, radiographs are still used as first-line studies to evaluate patients with possible fractures. Therefore, interpretation competence should constantly be evaluated. Interpretation errors in radiographs are frequently related to worse patient outcomes. There are still limited data on the diagnostic performance in MSK radiograph interpretation between specialists and residents, especially with regard to time of day and subtle vs. obvious findings. In this study, we evaluated only different upper and lower MSK regions due to their frequency and the limited number of imaging outcomes (e.g., fracture or no fracture).

The purpose of this study was to determine radiology specialists’ and residents’ performance in radiograph interpretation and the rate of discrepancy between them. We hypothesized that (1) radiology specialists’ performance is superior compared to residents’ performance, (2) residents have more missed findings in subtle radiology findings compared to specialists, and (3) missed findings increase during evening and night.

## 2. Materials and Methods

This retrospective cross-sectional study received ethical approval from the Ethics Committee of the University of Turku (ETMK Dnro: 38/1801/2020). This study complied with the Declaration of Helsinki and was performed according to ethics committee approval. Because of the retrospective nature of the study, need for informed consent was waived by the Ethics Committee of the Hospital District of Southwest Finland.

This retrospective study reviewed appendicular radiographs (N = 1006) interpreted by radiology specialists (*n* = 506) and residents (*n* = 500) between 2018 and 2021. This type of study design allowed us to collect the reports at one study point and was less time-consuming than a longitudinal or prospective study design. Different MSK body parts were included and the same amount of patient cases were included in every MSK region for both radiology specialists and trainees. Cases were selected with the following inclusion criteria: (a) trauma indication, (b) original radiology report made by either radiology specialists or residents, and (c) primary radiographs. The exclusion criteria were (a) non-trauma indication, (b) no original report found in PACS system, and (c) control study. All radiographs were interpreted by two subspecialty-level MSK radiologists with 20 and 25 years of experience. Double (dual) reading was used, which has been shown to be an effective but also time-consuming way of finding discrepancies in radiology reports [15]. The radiologists did not know the original report or whether the original report was made by radiology specialists or residents. Consensus between the two radiologists was evaluated against the original report. All radiographs were viewed in a picture archiving and communication system and with diagnostic monitors. To improve the generalizability of the results, data from various imaging devices were included.

Interpretation error was defined as disagreement between the original report and the two subspecialty-level MSK radiologists. In the case of interpretation error, it was evaluated and subcategorized. In addition, interpretation errors and their implications for patient treatment were classified based on the severity of the interpretation error. Implications were classified based on the consensus of the two subspecialty-level MSK radiologists as follows: Grade 1, no clinical importance; Grade 2, unable to know whether the error had clinical importance; and Grade 3, clear clinical effect on patient treatment. In addition, all abnormal radiographs were labeled as being subtle (*n* = 103) or obvious (*n* = 310) based on the two subspecialty-level MSK radiologists’ consensus.

Patient age, sex, time of interpretation, and date of interpretation were recorded. Data were collected and managed using REDCap (Research Electronic Data Capture) electronic data capture tools hosted at Turku University.

Patients were divided into three age groups (Table 1) to represent pediatric (1–16), adult (17–64) and elderly (>65). There were no statistically significant differences between patient age groups (*p* = 0.66) or sex (*p* = 0.53) when radiology specialists’ and residents’ interpretations were compared. In addition, time of interpretation was classified to present morning, evening, and night shifts.

Categorical variables were summarized with counts and percentages and continuous age with means together with range. Associations between two categorical variables were evaluated with chi-squared or Fisher’s exact test (Monte Carlo simulation used if needed). *p*-values less than 0.05 (two-tailed) were considered statistically significant. Sensitivity, specificity, positive predictive value (PPV), and negative predictive value (NPV) were calculated together with their 95% confidence intervals (CIs).

The data analysis for this paper was generated using SAS software version 9.4 for Windows (SAS Institute Inc., Cary, NC, USA).

## 3. Results

### 3.1. Overall Findings

Of the 1006 radiographs, 41% were abnormal. In total, 67 radiographic findings were missed (6.7%) and 32 findings were overcalled (3.2%). Among the missed fractures, 18% were found in children, 60% in adults, and 22% in elderly. Among the overcalls, 28.1% were found in children, 50% in adults, and 21.9% in elderly. The most common reason for interpretation error was fracture (58%). Interpretation error was most likely to happen in wrist (18%) or foot (17%) interpretations.

Different MSK regions had different rates of subtle and obvious radiographic findings (*p* = 0.001). Most subtle findings were found in the elbow (31%) and wrist (30%). Subtle radiographic findings occurred most often at 3 p.m.–4 p.m. (44%), 5 p.m.–6 p.m. (38%), and 9 p.m.–10 p.m. (38%). Figure 1 shows the distribution between morning, evening, and night shifts in interpretation errors, subtle and obvious findings, and abnormal radiographs.

There were no statistically significant differences between errors made in morning, evening or night shifts (*p* = 0.57) (Table 2). Radiology specialists were better at correctly diagnosing radiographs during the evening and nighttime compared to radiology residents (93% vs. 87%), but there were no statistically significant differences. Error rates did increase for radiology specialists during 7–8 a.m., 11–12 a.m., 15–17 p.m. and 23–00 p.m. The highest error rates for radiology residents were found during 1–4 a.m., 6–7 a.m. and 16–18 p.m. There were no statistically significant differences between misses, overcalls and weekdays (*p* = 0.31). Most misses were made on either Monday (22%) or Saturday (22%). In addition, most overcalls were made on Friday (28%).

### 3.2. Discrepancies between Radiology Specialists and Residents

No statistically significant differences (*p* = 0.44) were found in the interpretation errors between the radiology specialists and the residents. The radiology specialists missed 5.7% of the findings, while the residents missed 7.6%. On the other hand, the radiology specialists made 2.8% of the overcalls and the residents made 3.6% of the overcalls. The sensitivity, specificity, positive predictive value, and negative predictive value were 0.86, 0.92, 0.88, and 0.91, respectively (Table 3). Patient age was similar (*p* = 0.29) in the correct diagnosis group and in the interpretation error group. However, there were variations in competence between the different MSK regions and radiology specialists or residents.

Diagnostic accuracy in the different MSK regions showed a wide range of variation (Table 4). The highest sensitivity (0.97), specificity (0.95), negative predictive value (0.97), and positive predictive value (0.95) were found in the pelvis interpretation, while the lowest sensitivity (0.82), specificity (0.83), negative predictive value (0.80), and positive predictive value (0.85) were found in the wrist interpretation. Overall, the lowest sensitivity (0.78) was found in the foot interpretation. For the shoulder, the radiology specialists made the correct diagnoses in 95% of the cases, compared to 83% by the residents; for the knee, the radiology specialists made the correct diagnoses in 89% of the cases, compared to 97% by the residents. However, there were no statistically significant differences between the radiology specialists and the residents in the different MSK regions.

Radiology specialists interpreted more radiographs as having subtle findings compared to residents (*p* = 0.04). Different age groups did not differ (*p* = 0.89) between subtle or obvious cases. Radiology specialists missed correct diagnoses in subtle and obvious radiographs in 33% and 4.9%, respectively. In contrast, residents missed correct diagnoses in subtle (Figure 2) and obvious (Figure 3) radiographs in 51% and 8.4%, respectively.

From all the missed findings in the radiographs, 70% (*n* = 44) were interpreted as having an impact on patient care (*p* = 0.02), but this did not differ between the radiology specialists and the residents. Findings missed by the radiology specialists (Figure 4 and Figure 5) affected patient care in 71% of cases and overcalls in 31% of cases. Findings missed by the residents (Figure 6) affected patient care in 69% of cases and overcalls in 47% of cases. From all the overcalls in the radiographs, 40% (*n* = 12) seemed to have an impact on patient care. The most common impact on patient care was a lack of the necessary control study (40%), followed by an unnecessary control study (14%). Interpretation error rarely led to unnecessary operative treatment (1%).

## 4. Discussion

### 4.1. Overall Findings

We found similar rates of misses and overcalls in the reading of the radiographs between the radiology specialists and the residents, with both groups having lower sensitivity compared to specificity, yet there were differences in competence among the different MSK regions. Neither day nor time of the day showed statistically significant differences in interpretation competence. These results highlight that there are no major differences between the radiology specialists and the residents in MSK radiograph interpretation. However, there are MSK regions that need more attention in the future regarding competence in radiograph interpretation. This will have direct implications for resident training programs. Importantly, there were no statistically significant group differences in the age distribution between the resident and specialist groups, suggesting that the main conclusions are not biased by age.

For the upper and lower extremities, we found a sensitivity of 0.86 and specificity of 0.92, which are lower than reported in the previous studies [16]. In contrast to the previous studies [16,17], we did not find any statistically significant increase in the radiology specialist or resident interpretation errors for the evening or night shifts compared to the daytime shifts. However, we did find that the residents, who can be more prone to fatigue-related errors [18,19], made more interpretation errors during the night shift compared to the morning or evening shift. The radiology specialists are also prone to fatigue-related problems in interpretation [17] and, in this study, we found that 18% of missed diagnoses occurred between 15:00 and 17:00, which highlights the fatigue-related errors in interpretation. Most missed diagnoses in this study were related to missed fractures, similar to the previous studies [20,21,22]. The prevalence of abnormality in our study was 41%, which is in line with prevalence in clinical practice [23] and does not overestimate the ability to detect abnormal cases [24].

### 4.2. Discrepancies between Radiology Specialists and Residents

We found that the overall interpretation errors for radiology specialists and residents varied from 0 to 10% and 0 to 12%, respectively, showing slightly lower competence levels compared to previous studies [1,7,21,25,26,27,28]. Earlier studies show that when evaluated with normal and abnormal cases, interpretation errors for radiology specialists range from 0.65% [1] to 5% [29,30]. There are differences between individual radiology specialists’ interpretation competence, which can increase interpretation errors even to 8% [31]. One of the largest studies showed a radiology specialist interpretation error rate between 3% and 4% [1].

We did not find any statistically significant differences between the radiology specialists and the residents, which is in contrast to the previous studies, where the radiology specialists showed better diagnostic accuracy compared to the residents (*p* = 0.02) [32]. However, there are also studies showing no significant differences between the radiology specialists and the residents [1,20,25]. In addition, we did not find statistically significant differences in the interpretation of subtle or obvious radiology findings, in contrast to the previous studies [32]. In this study, the radiology specialists had higher rates of detection and higher diagnostic accuracy for subtle findings compared to the residents, which is consistent with the previous studies [18]. Because we excluded reports initially signed by both a trainee and a specialist (a signal of consultation), the potential bias from specialists affecting trainee reports is probably low. In addition, we did not find statistically significant differences between the radiology specialists and residents in different MSK regions, as in the previous studies [33]. In the previous studies [16,30], ankle interpretation showed the highest sensitivity (0.98) and specificity (0.95). In this study, the ankle sensitivity (0.83) and specificity (0.93) were lower. Furthermore, in this study, sensitivity was lower compared to specificity in all the MSK regions except the pelvis. This is well recognized in the field of radiology and can be related to litigation in missed findings [34].

Diagnostic accuracy in the wrist had the lowest sensitivity and specificity among the MSK regions. This is worrying because the wrist is the most often injured MSK region [35,36], and missed findings can lead to complications such as nonunion, osteonecrosis, and osteoarthritis [6]. The radiology specialists and residents had the same miss rate, with 9.5% and 9.7%, respectively, but the radiology specialists had fewer overcalls compared to the residents, with 3.6% and 8.1%, respectively. These miss and overcall rates in the wrist are higher than reported in the previous studies [37]. Foot injuries are also very common, and diagnostic accuracy can have serious implications on patient care [38]. In our study, foot interpretation showed the lowest sensitivity and specificity in the lower extremity. These findings should prompt radiology departments to pay special attention to these MSK regions in resident training. We found that most interpretation errors affected patient care, regardless of whether the radiograph was interpreted by a radiology specialist or resident.

### 4.3. Limitations

First, due to the retrospective nature of the study, we were unable to verify the level of clinical competence of the radiology specialist (e.g., years in practice), or the resident (e.g., year of residency). However, we might reasonably assume that every radiology specialist or resident has the required clinical competence when they dictate radiological reports for guiding patient treatment. Second, there is a possibility of undetected selection bias. Different types of fracture tend to occur during different times of the year in Finland. To diminish this selection bias, data collection spanned several time periods. Finally, follow-up studies were not obtained to verify the possible missed fractures unless the patient had had follow-up assessment at the same hospital and it could be found in the PACS. Our gold standard in this study was a consensus of two MSK radiology specialists, and possible errors in their interpretations potentially affect the results of this study also.

## 5. Conclusions

In conclusion, this study found a lack of major discrepancies between radiology specialists and residents in radiograph interpretation, although there were differences between MSK regions and in subtle or obvious radiographic findings. In this study, the interpretation of pelvic imaging yielded the most notable outcomes in terms of sensitivity, specificity, negative predictive value (NPV), and positive predictive value (PPV), whereas the interpretation of wrist radiographs demonstrated the most modest results in these performance metrics. Moreover, it is worth noting that no statistically significant distinctions were observed between the interpretations made by radiology specialists and trainees during evening or night shifts, despite radiology specialists showing a reduced incidence of interpretational errors. In addition, missed findings found in this study often affected patient treatment. Finally, there are MSK regions where the sensitivity or specificity are below 90%, and these should raise concerns and highlight the need for double reading and be taken into consideration in radiology education. Further prospective studies are needed in these specific MSK regions. In addition, future studies where artificial image interpretation is compared between radiology specialists and residents could be undertaken to highlight possible differences.

## Figures and Tables

**Figure 1 diagnostics-13-03207-f001:**
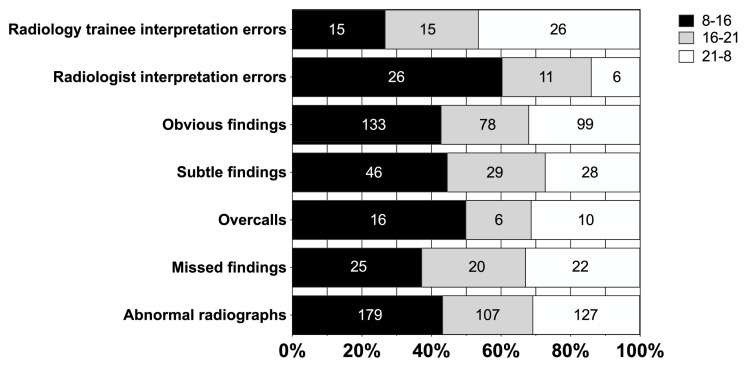
Total number and percentage of abnormal radiographs, missed diagnoses, and overcalls; subtle and obvious findings presented in three different timeframes.

**Figure 2 diagnostics-13-03207-f002:**
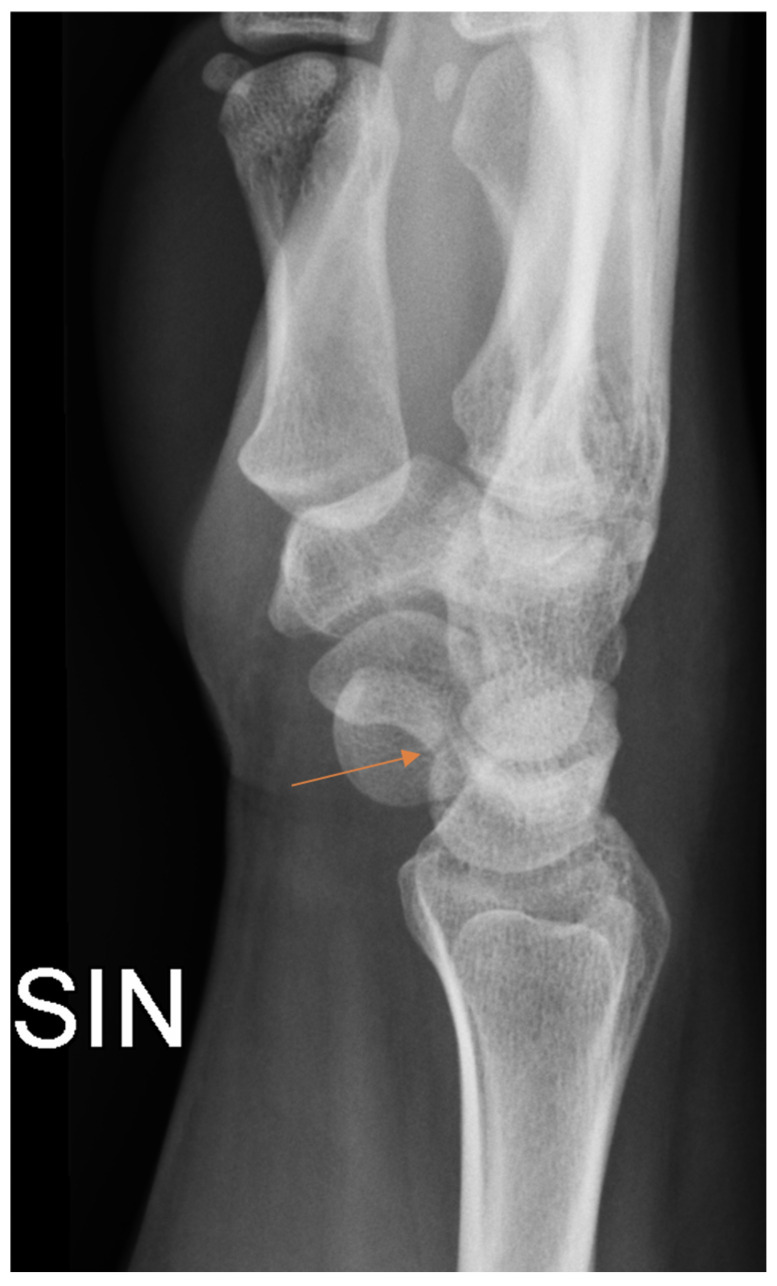
Subtle radiographic finding in patient with scaphoid fracture (arrow) that was initially missed by the resident.

**Figure 3 diagnostics-13-03207-f003:**
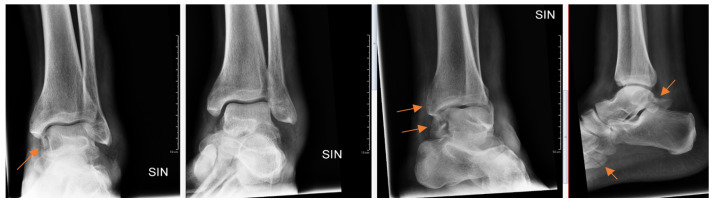
Patient with ankle trauma. Multiple obvious findings (arrows) in radiographs that were all missed by the resident.

**Figure 4 diagnostics-13-03207-f004:**
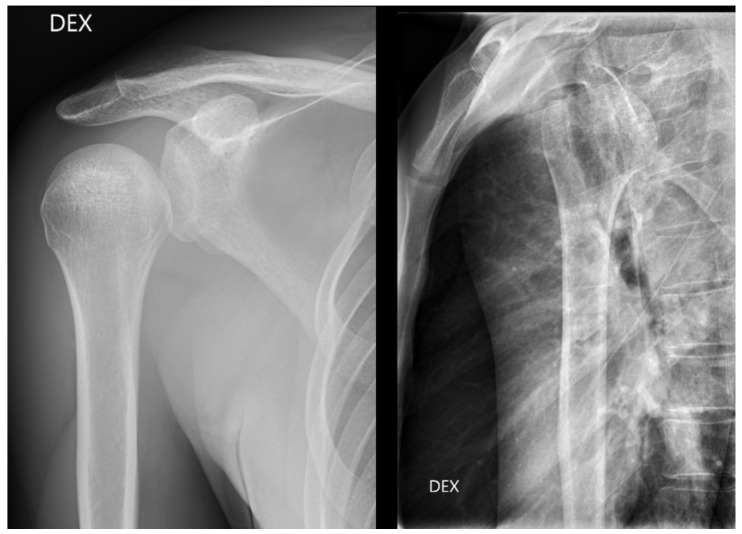
Posterior dislocation initially missed by the radiology specialist. The treating physician later suspected GH dislocation on clinical inspection, and a CT was ordered where posterior dislocation was detected.

**Figure 5 diagnostics-13-03207-f005:**
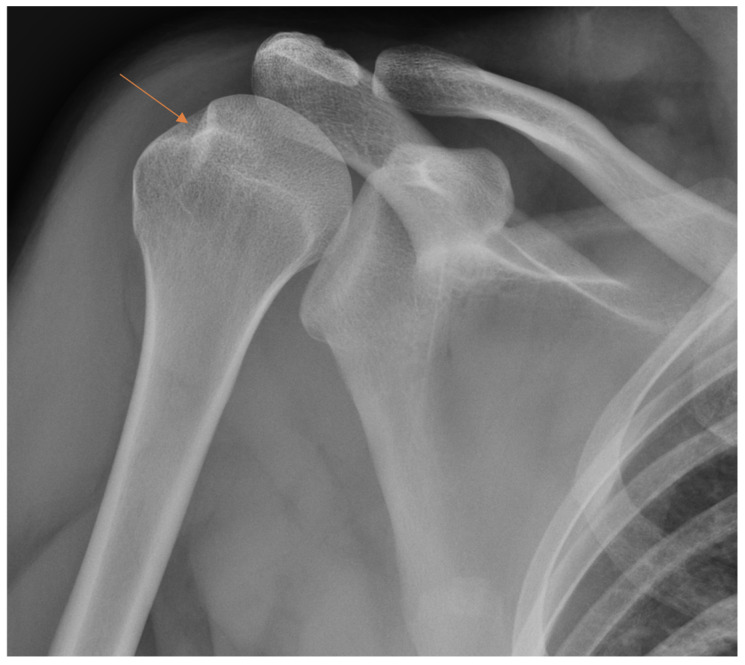
Patient with anterior shoulder dislocation. The radiology specialist missed a Hill–Sachs lesion (arrow) that resulted in delay in patient treatment.

**Figure 6 diagnostics-13-03207-f006:**
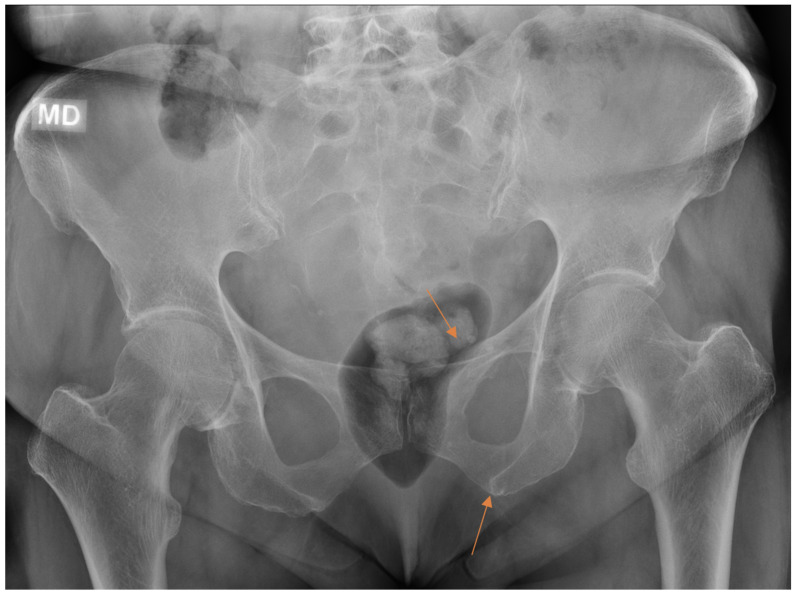
Patient with pelvic trauma radiographs. Two findings (arrows) initially missed by the resident were later revealed on CT done for other indications.

**Table 1 diagnostics-13-03207-t001:** Patient demographics in different subsets.

Patient Demographics	Radiology Specialists’ Evaluation (*n* = 506)	Radiology Residents’ Evaluation (*n* = 500)	Total (N = 1006)
Age (y)			
Mean			45.4 (1–99)
1–16	88 (17.4%)	92 (18.4%)	11.6 (1–16)
17–64	255 (50.4%)	260 (52.0%)	36.7 (17–64)
>65	163 (32.2%)	148 (29.6%)	79.4 (65–99)
Sex			
Male	218 (43.1%)	226 (45.2%)	444 (44.1%)
Female	288 (56.9%)	274 (54.8%)	562 (55.9%)

**Table 2 diagnostics-13-03207-t002:** Diagnostic accuracy of radiology specialists’ and residents’ interpretations at different times.

	Sensitivity	Specificity	PPV	NPV
Daytime (8:00–16:00) (*n* = 444)	0.89 (0.85–0.94)	0.92 (0.89–0.95)	0.88 (0.84–0.93)	0.93 (0.90–0.96)
Daytime (8:00–16:00) Radiology specialist (*n* = 287)	0.89 (0.83–0.94)	0.92 (0.88–0.96)	0.88 (0.82–0.94)	0.92 (0.89–0.96)
Daytime (8:00–16:00) Radiology resident (*n* = 157)	0.91 (0.84–0.98)	0.92 (0.87–0.98)	0.90 (0.82–0.97)	0.93 (0.88–0.98)
Evening and night (16:01–07:59) (*n* = 562)	0.84 (0.79–0.88)	0.91 (0.88–0.94)	0.87 (0.83–0.91)	0.89 (0.85–0.92)
Evening and night (16:01–07:59) Radiology specialist (*n* = 219)	0.85 (0.77–0.92)	0.94 (0.90–0.98)	0.91 (0.84–0.97)	0.90 (0.84–0.95)
Evening and night (16:01–07:59) Radiology resident (*n* = 343)	0.83 (0.77–0.89)	0.90 (0.85–0.94)	0.85 (0.79–0.91)	0.88 (0.84–0.93)

PPV = positive predictive value, NPV = negative predictive value.

**Table 3 diagnostics-13-03207-t003:** Diagnostic accuracy of radiology specialists’ and residents’ trainee interpretations.

	Sensitivity	Specificity	PPV	NPV
UE and LE (*n* = 1006)	0.86 (0.83–0.90)	0.92 (0.89–0.94)	0.88 (0.84–0.91)	0.91 (0.88–0.93)
Radiology specialist (*n* = 506)	0.87 (0.82–0.91)	0.93 (0.90–0.96)	0.89 (0.85–0.93)	0.91 (0.88–0.94)
Radiology resident (*n* = 500)	0.86 (0.81–0.90)	0.90 (0.87–0.94)	0.86 (0.82–0.91)	0.90 (0.86–0.93)
UE (*n* = 495)	0.86 (0.81–0.90)	0.91 (0.87–0.94)	0.89 (0.84–0.93)	0.88 (0.85–0.92)
UE Radiology specialist (*n* = 249)	0.86 (0.80–0.93)	0.95 (0.91–0.99)	0.93 (0.88–0.98)	0.90 (0.85–0.95)
UE Radiology resident (*n* = 246)	0.85 (0.79–0.92)	0.86 (0.80–0.92)	0.85 (0.78–0.91)	0.87 (0.81–0.93)
LE (*n* = 511)	0.87 (0.82–0.92)	0.92 (0.89–0.95)	0.87 (0.82–0.92)	0.92 (0.89–0.95)
LE Radiology specialist (*n* = 257)	0.87 (0.80–0.94)	0.91 (0.86–0.95)	0.85 (0.77–0.92)	0.93 (0.88–0.97)
LE Radiology resident (*n* = 254)	0.86 (0.79–0.93)	0.94 (0.90–0.98)	0.89 (0.82–0.95)	0.92 (0.88–0.96)

PPV = positive predictive value, NPV = negative predictive value.

**Table 4 diagnostics-13-03207-t004:** Diagnostic accuracy of radiology specialists’ and residents’ interpretations at different MSK regions.

	Sensitivity	Specificity	PPV	NPV
Hand (*n* = 121)	0.82 (0.71–0.93)	0.94 (0.89–1.00)	0.91 (0.83–0.99)	0.88 (0.81–0.95)
Wrist (*n* = 125)	0.82 (0.73–0.91)	0.83 (0.73–0.92)	0.85 (0.76–0.93)	0.80 (0.70–0.90)
Elbow (*n* = 129)	0.92 (0.84–1.00)	0.94 (0.88–0.99)	0.90 (0.82–0.98)	0.95 (0.90–1.00)
Shoulder (*n* = 120)	0.88 (0.80–0.96)	0.90 (0.82–0.98)	0.90 (0.82–0.98)	0.89 (0.81–0.97)
Pelvis (*n* = 123)	0.97 (0.92–1.00)	0.95 (0.90–1.00)	0.95 (0.89–1.00)	0.97 (0.93–1.00)
Knee (*n* = 127)	0.88 (0.75–1.00)	0.92 (0.87–0.97)	0.73 (0.58–0.89)	0.97 (0.93–1.00)
Ankle (*n* = 136)	0.83 (0.72–0.93)	0.93 (0.87–0.98)	0.88 (0.78–0.97)	0.90 (0.83–0.96)
Foot (*n* = 125)	0.78 (0.67–0.90)	0.89 (0.82–0.96)	0.83 (0.73–0.94)	0.86 (0.78–0.94)

PPV = positive predictive value, NPV = negative predictive value.

## Data Availability

The data presented in this study are available on request from the corresponding author.

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
