# Peer review of "Discrepancies between Radiology Specialists and Residents in Fracture Detection from Musculoskeletal Radiographs"

_diagnostics, 2023, doi:10.3390/diagnostics13203207_

Round 1

Reviewer 1 Report

The study is an interesting one. I have only a few suggestions.

1. Major contribution of the study may be highlighted in the introduction section.

2. Research questions may be framed.

3. Knowledge gaps may be elaborated more.

4. A comparison of the study with the state-of-the-art studies may be incorporated.

5. The state-of-the-art studies may be reviewed with pros and cons in a tabular form.

6. Future research directions may be added in the conclusion.

Reviewer 2 Report

This study reports the discrepancies between radiology specialists and residents for reading radiographs. The results include useful information, however, there are unclear points in the manuscript. They should be revised.

Title

This study reports on the radiographs of MSK only. It might be better to include the word “MSK” in the title.

1.     Introduction

The reasons why only MSK has been selected in this study should be described more clearly.

There have been similar studies on the comparisons between specialists and persons who do not have enough experiences in this field. The authors compared this study and previous reports. However, the differences between them are not clear. More explanations would be necessary.

As the rooms of reading radiographs are generally not bright, so the purpose to compare daytime, evening and night might be to investigate the contributions of fatigues. However, the contents and densities of their works could differ between two groups. Could this factor be ignored?

2.     Materials and Methods

The number of the specialists and residents were 506 and 500, respectively. As these number are large, the characteristics of them might be not homogeneous. If so, doesn’t this affect the results?

The actual qualities of diagnoses would depend not only on the differences of experiences of readers but also on the machines or imaging systems. Have there been any differences on the kinds of medical equipment among medical facilities attended in this study? If there are any findings, they should be described.

3.     Results

The authors show the missed findings points. It would be better to describe the reasons of the missing if possible.

4.     Discussion

The patients were classified into three age groups. Are there any findings of the results in terms of the age groups, such as skeleton size, shape or bone density?

Figures

Figure 1: The numbers and words in Figure 1 are so small to read.

Reviewer 3 Report

The purpose of the study is to compare the competence in appendicular trauma radiograph image interpretation between radiology specialists and residents.

Abstract

I suggest to:

- divide the abstract in section

- improve the explanation of results

- improve the conclusions

Introduction

I suggest to:

 summarize this section

Material and Method

I suggest to:

-  improve patients’ selection (there were differences between two groups in age, sex, region evaluated?)

-  improve the explanation of statistical methods

Results, Discussion and Conclusion

I suggest to:

-  include the p value for your statistical analysis

- from the results seems that young radiologists obtained similar results than the older radiologists but from the conclusions it doesn't seem like it.

Round 2

Reviewer 3 Report

The authors sufficiently improve their article.